# Effects of Music-Based Interventions on Motor and Non-Motor Symptoms in Patients with Parkinson’s Disease: A Systematic Review and Meta-Analysis

**DOI:** 10.3390/ijerph20021046

**Published:** 2023-01-06

**Authors:** Hyunjung Lee, Bumsuk Ko

**Affiliations:** 1Department of Music Therapy, Graduate School, Ewha Womans University, Seoul 03760, Republic of Korea; 2Department of Music, Graduate School, Hansei University, Gunpo 15852, Republic of Korea

**Keywords:** Parkinson’s disease, motor symptoms, non-motor symptoms, music therapy, meta-analysis

## Abstract

This systematic review and meta-analysis examined previous studies on music-based interventions for individuals with Parkinson’s disease (PD). The effectiveness of the interventions on various motor and non-motor outcomes was evaluated. This review was conducted by searching PubMed, CINAHL, PsycINFO, and Cochrane Library CENTRAL prior to June 2022 for randomized controlled trial (RCT) and controlled clinical trial (CCT) studies published in English. Data were expressed as weighted/standardized mean difference (MD/SMD) with 95% confidence intervals (CI). I^2^ index was used for heterogeneity. The initial search identified 745 studies, and 13 studies involving 417 participants with PD which met the inclusion criteria included in this review. The results of the meta-analysis revealed that music-based interventions can significantly improve walking velocity (MD = 0.12, 95% CI = 0.07~0.16, *p* < 0.00001), stride length (MD = 0.04, 95% CI = 0.02~0.07, *p* = 0.002), and mobility (MD = −1.05, 95% CI = −1.53~−0.57, *p* < 0.0001). However, the results did not support significant effects for music-based interventions on cadence (MD = 3.21, 95% CI = −4.15~10.57, *p* = 0.39), cognitive flexibility (MD = 20.91, 95% CI = −10.62~52.44, *p* = 0.19), inhibition (SMD = 0.07, 95% CI = −0.40~0.55, *p* = 0.76), and quality of life (SMD = −0.68, 95% CI= −1.68~0.32, *p* = 0.18). The findings suggest that music-based interventions are effective for the improvement of some motor symptoms, but evidence for non-motor symptoms is limited. Further high-quality studies with a larger sample size are required to obtain the robust effects of music-based interventions on various outcomes among patients with PD.

## 1. Introduction

Parkinson’s disease (PD) is the second most common age-related neurodegenerative disorder after Alzheimer’s disease (AD), affecting more than 1% of the global population over the age of 65 years and 5% of individuals up to 85 years of age [1]. PD is caused by a loss of dopamine production, leading to changes in the central nervous system that impair the neural networks including the basal ganglia and supplementary motor areas [2]. It is characterized by gradual decline in not only motor functions but also non-motor functions such as cognitive and emotional disturbances [3]. Alongside the well-recognized motor features of PD including resting tremor, muscular rigidity, bradykinesia, gait impairment, freezing, and postural instability [4,5], speech and communication skills are often affected [6] and cognitive impairments, depression, and anxiety are also prevalent in PD patients [7,8]. Consequently, all these symptoms could substantially affect quality of life and well-being in individuals with PD [9].

Current dominant forms of treatment for PD involve multiple drug therapy and deep brain stimulation. They provide significant relief but have serious side effects and are expensive [10,11]. Moreover, long-term medical treatment may be associated with treatment-resistant symptoms and motor complications including dyskinesia, fluctuations, and choreoathetosis [12,13]. Thus, therapeutic strategies that limit side effects, reduce treatment costs, and target multiple symptoms of PD are of increasing interest in non-pharmacological interventions in PD [14].

Music-based interventions have been used for many years as an alternative therapeutic medium in patients with PD [9,15,16]. Music can engage and modulate broad regions in the brain which are involved in the perception and regulation of behavior, movement, mood, and cognitive functions, and thus it has been proven useful in neurologic rehabilitation [17,18]. Music activities can be performed in an active way, such as playing an instrument or singing, or in a passive way, such as listening [16,17,18,19]. With diverse types of approaches, music has been successfully used to address both motor and non-motor symptoms in PD. Regarding motor symptoms, a great number of studies have shown that music as rhythmic auditory stimuli can improve gait-related movements, functional mobility, and balance, as well as freezing of gait and falls of gait in PD [20,21,22,23,24,25]. According to Calabrò et al. [26], rhythmic auditory stimulation (RAS) can compensate for the loss of automatic and rhythmic movements. RAS is one of the well-known techniques of neurologic music therapy (NMT), which aims to develop and improve gait pattern through rhythmic auditory cues using metronome beats or strong beats embedded in music [27]. A study by Calabrò et al. suggests that the use of rhythmic auditory cueing during gait training facilitates the restoration of internal synchronization mechanisms that generate and control motor rhythmicity, thus improving gait-related performance [26].

On the other hand, research in the non-motor domain has recently started to investigate the effect of music-based interventions on various outcomes including cognitive functions [19,28,29,30], speech and communication skills [31,32,33,34,35], and emotional and psychosocial symptoms [16,30,32,36,37,38,39,40,41]. Spina et al. [30] conducted a pilot study to investigate the effect of music activities on cognition in patients with PD. The results demonstrated an improvement of executive functions such as cognitive flexibility, processing speed, attention, and working memory, which suggests that music-based interventions could improve frontal function by serving as a training ground for these cognitive skills. In a study by Tamplin et al. [33], it was considered that the activity of singing shares many of the neural networks and structural mechanisms used during speech and therefore has the potential for therapeutic applications to address speech and communication disorders. Music activities such as listening to music, singing, and improvisation with music can also impact the limbic system and neurochemical circuits (the reward system) and provide positive effects on emotional and psychosocial symptoms in patients with PD [41].

Currently, several systematic reviews and meta-analyses have analyzed the effects of music-based interventions in PD, suggesting that these interventions may be potentially beneficial in improving motor and non-motor symptoms among people with PD [9,42,43,44,45,46,47]. In some review studies, the use of rhythmic timing has been extended to RAS studies [43,44] and music-based movement therapy [44,45], and the results strongly support the effects of music-based interventions on motor functions in PD. A few review studies have investigated non-motor symptoms including cognitive functions, depression, and quality of life [9,44,45,47] and the results among those studies have been controversial regarding the effectiveness of music interventions on non-motor symptoms. A recent systematic review study investigated the potential benefit of active group-based performing arts interventions on both motor and non-motor symptoms in patients with PD [47]. Fifty-six studies were analyzed and covered four active arts modalities such as dance, singing, music therapy, and theatre. The outcomes included motor, communication and speech functions, and cognitive status, as well as quality of life in PD. The evidence suggests that performing arts including music-based interventions performed in an active way may be an effective therapeutic medium for patients with PD. The use of music-based interventions for the management of motor and non-motor symptoms in individuals with PD has been documented previously. However, previous systematic reviews and meta-analyses were focused either on motor symptoms or non-motor symptoms or included a certain type of music activity as the intervention. To the best of our knowledge, to date, there have been no systematic reviews and meta-analyses targeting both motor and non-motor symptoms in individuals with PD, applying active and passive music activities as interventions. Accordingly, the aim of this study was to conduct a critical review of the effectiveness of various types of music-based interventions on motor and non-motor symptoms in patients with PD.

## 2. Materials and Methods

This systematic review and meta-analysis was conducted in accordance with the guidelines outlined in preferred reporting items for systematic reviews and meta-analyses (PRISMA) 2020 [48] and the protocol of the present study was registered in the International Prospective Register of Systematic Reviews (PROSPERO; No. CRD42022372507).

### 2.1. Data Sources and Search Strategy

Two researchers independently searched four databases, CINAHL, Cochrane Library CENTRAL, PsycINFO, and PubMed for all articles published prior to 30 June 2022. The researchers also explored music therapy journals including the Journal of Music Therapy, Music Therapy Perspectives, and the Nordic Journal of Music Therapy. The language was English only and the combinations of search words were “Parkinson” AND “Music” AND (“Intervention” OR “Therapy” OR “Treatment”).

### 2.2. Eligibility Criteria

The following inclusion criteria were guided by the PICOS strategy [49].
Participants: Patients of any age, gender, or disease stage who have been diagnosed with PD. However, studies including atypical PD and other diagnoses (e.g., Alzheimer’s Disease) were excluded.Interventions and comparisons: The inclusion criteria for the interventions in the experimental group were any music-based intervention or other interventions combined with music, but the studies which used music as a secondary medium were excluded. Other modalities of intervention including dance and movement therapy using music as a secondary medium of treatment were also excluded. Studies which examined the immediate effect of interventions were also excluded. The comparison groups included those who received non-music interventions, usual care, or no intervention. The intervention format (individual/group), setting (clinic/hospital/home), duration, and frequency were not limited.Outcomes: There was no limit to the dependent variable that music-based interventions affected. Therefore, the outcomes of interest were both motor (e.g., gait, mobility) and non-motor symptoms (e.g., executive function and quality of life). All outcomes were measured using validated tools including the timed up-and-go test (TUG), the trail making test (TMT) and the Stroop color and word test (SCWT).Study design: The study design included in this review were randomized controlled trials (RCTs) and controlled clinical trials (CCTs). This study was limited to articles published in English and peer-reviewed journals. The following types of research were ruled out: master’s theses, doctoral dissertations, conference papers, case studies, descriptive studies, literature research, and any review studies.

### 2.3. Study Selection and Data Extraction

After removing the duplicate articles, two researchers independently screened the titles and abstracts of all of the articles. Then, the researchers assessed the full-text articles and excluded those that did not meet the inclusion criteria. The final articles were selected through discussion between the researchers. Then, two researchers independently extracted data using a data extraction form, which included authors, publication year, country, sample size, sample description (i.e., age, gender, disease duration, and disease severity), and interventions, as well as outcomes. Disagreements encountered during the process were resolved through discussion and consensus. In cases of missing valuable data or information, valid data were obtained by contacting the corresponding authors of the articles. If data were still unavailable, then the article was not included in the meta-analysis but was included in the systematic review.

### 2.4. Risk-of-Bias Assessment

The risk of bias in the included studies was evaluated using the Cochrane risk-of-bias tool for randomized trials (RoB 2) for RCTs [50], and the risk of bias in non-randomized studies for interventions (ROBINS-I) for CCTs [51]. The scale of RoB 2 includes five domains: bias arising from the randomization process, bias due to deviations from intended interventions, bias due to missing outcome data, bias in measurement of the outcome, and bias in selection of the reported result. The overall judgement for each RCT study was classified as being “low”, “some concerns”, and “high”. Meanwhile, the tool of ROBINS-I includes seven domains: bias due to confounding, bias due to selection of participants, bias in classification of interventions, bias due to deviations from intended interventions, bias due to missing data, bias in measurement of outcomes, and bias in selection of the reported result. The overall risk of bias for each CCT study was rated as “low”, “moderate”, “serious”, and “critical”. Two reviewers completed a risk-of-bias assessment independently and they resolved discrepancies through discussion. In line with the guidance provided by Cochrane, studies at a high risk of bias according to ROB 2 and those at a critical risk of bias according to ROBINS-I were excluded in the meta-analysis, but were included in the systematic review [52].

### 2.5. Data Synthesis

The included studies that reported sufficient numerical data to calculate effect sizes were synthesized quantitatively in the meta-analysis. The meta-analyses were conducted on outcomes for which data from at least three studies could be synthesized. The data analysis was performed using the Review Manager 5.4.1 software provided by the Cochrane Collaboration and statistical significance was defined as a *p*-value < 0.05. The outcomes in this study were continuous variables, and the researchers calculated 95% confidence intervals (CIs) for each effect size. Data were pooled using mean difference (MD) for outcomes evaluated by the same scale and standardized mean difference (SMD) was calculated for outcomes evaluated using different scales. The heterogeneity of the study was expressed by I^2^, and the low-, moderate-, and high-degree heterogeneities were denoted by the I^2^ statistics of 25%, 50%, and 75%, respectively. A random-effects model was applied if I^2^ ≥ 50% and *p* < 0.1. Otherwise, a fixed-effects model was chosen. In addition, sensitivity analyses were performed to verify the robustness of the main findings. Visual inspection of funnel plots can be used to confirm publication bias if more than 10 studies meet the conditions of the meta-analysis, in accordance with the guidelines in the Cochrane handbook for systematic reviews of interventions [53].

## 3. Results

### 3.1. Study Selection

Initially, 745 articles were retrieved from the databases CINAHL, PsycINFO, PubMed, and Cochrane Library CENTRAL. One hundred eighty-nine articles were removed after screening out duplicate records. From the remaining 556 articles, 523 were excluded based on their titles and abstracts. Then, one additional duplicate study in a different title was removed. Thirty-two articles were assessed for their eligibility and 17 of those studies remained. After conducting a risk-of-bias assessment, four studies with a high and critical risk of bias were excluded. Ultimately, a total of 13 articles met the inclusion criteria for a systematic review, leaving 11 articles for the meta-analysis. At this stage, two studies were excluded due to insufficient data for meta-analysis. In addition, 108 articles were selected from the Journal of Music Therapy, Music Therapy Perspectives, and the Nordic Journal of Music therapy, but none of those articles satisfied the eligibility criteria of this study. The detailed flow diagram of the literature search, screening, and selection is shown in Figure 1.

### 3.2. Summary of Risk of Bias

Figure 2 shows the risk-of-bias results for the RCT studies. Among 14 studies, 1 study was classified as “low risk” [26], 9 studies were classified as “some concerns” [16,27,30,36,39,54,55,56,57], and 4 studies were classified as “high risk” [19,38,58,59]. The most common risk of bias was missing outcome data due to the reporting of dropouts. Four studies were at high risk of bias for this reason [19,38,58,59]. One study described the difference in baseline outcomes between groups [58] and the other study was identified by deviation from the intended interventions, as participants were not blinded to the intervention allocation [38]. Meanwhile, Figure 3 displays the risk-of-bias results for the CCT studies. Among three studies, one study was at a moderate risk of bias [29] and other two studies were at a serious risk of bias [28,37]. One study reported that the selection of participants might impact the outcome [28] and the other study selectively reported outcomes which can lead to a high risk of bias [37]. None of studies were at a critical risk of bias among the CCT studies. Detailed summaries of risk of bias are shown in Figure 2 and Figure 3.

### 3.3. Study Characteristics

A total of 13 studies included in the review were published between 1996 and 2021. Four studies were carried out in Italy and three studies were conducted in the United States. The remaining six studies were performed in Canada, India, Korea, Poland, Romania, and Thailand. Ten studies were RCTs, and three studies were CCTs.

#### 3.3.1. Characteristics of Participants

Table 1 summarizes the differences in the fundamental information of the participants in the music-based intervention group and the control group. A total of 417 participants with PD were involved in the study and the range of sample sizes was from 11 to 55 with a mean age between 61.6 and 74 years. Ten studies included both genders, with males accounting for 56% (n = 187), while three studies did not specify the gender information [30,56,57]. The range of the mean duration of PD was between 2 and 15.4 years and five studies did not report this information [27,29,36,37,54]. For the disease severity, unified Parkinson’s disease rating scale (UPDRS) and/or Hoehn and Yahr scores were used and the disease stage of PD was between 1 and 3, assessed by the Hoehn and Yahr scores in most studies. One study did not provide Hoehn and Yahr scores, but rather UPDRS scores, with them reporting mild disability for participants’ disease severity [30].

#### 3.3.2. Characteristics of Interventions

Several music-based interventions in the studies mainly provided physical activities. Five studies provided RAS for gait rehabilitation [26,27,54,55,56]. Specifically, among those studies using RAS training, one study [27] provided RAS combined with other neurologic music therapy (NMT) techniques, such as therapeutic instrumental music performance (TIMP) and patterned sensory enhancement (PSE), in order to facilitate the participants’ gait pattern and balance. In three other studies, RAS was applied with treadmill training [26,54,56]. Two studies provided conventional physical therapy combined with listening to music [36,39] and one study provided music-contingent stepping training [28]. Some studies addressed non-motor components such as the participants’ cognitive and psychosocial symptoms [16,29,30,36,39,57]. Two of the studies provided various musical activities including listening, singing, playing, and dancing [16,30], while the interventions in other studies focused on a form of instrumental playing such as piano and drum performance [36,37,57]. The duration, frequency, and time of the interventions varied among the studies. The duration of the interventions was concentrated on 10 days to 24 weeks, with the range of frequency from 1 to 7 times per week while the intervention times varied from 10 min to 3 *h*. As for the type of intervention, seven studies provided group sessions [16,29,30,36,37,39,57] and six studies provided individual sessions [26,27,28,54,55,56]. Individual sessions were provided primarily for gait-related training. Active control groups providing non-musical interventions including treadmill gait training and conventional physical therapy were used for comparison in most studies [16,26,28,36,39,54,55,56,57]. On the other hand, some studies made use of comparison groups that implemented daily activity [27,55], or no intervention [29,30,37]. Most of the research focused on motor outcomes, including general motor function (i.e., mobility, balance, stability, freezing of gait, finger dexterity, etc.) and gait parameters (i.e., cadence, velocity, stride length, step width, stance phase, swing phase, etc.). These motor functions were assessed by a wide range of methods. However, there were certain assessment tools that were used frequently, such as the timed up-and-go test (TUG) and the motor subscale of the unified Parkinson’s disease rating scale (UPDRS), either in its original [16] or revised version [29]. Other methods included the Berg balance scale (BBS), the nine-hole peg test (9HPT), the 10m walking test (10MWT), the functional gait assessment (FGA), and the freezing of gait questionnaire (FOG-Q). One study used electromyography (EMG) in order to measure the variability, symmetry, and timing of the muscle activation period [55]. Non-motor outcomes in the studies included attention, memory, visuospatial function, language, executive function, general cognitive function, quality of life, and mental health. The evaluation methods applied to the non-motor symptoms that were featured most commonly included the Stroop color and word test (SCWT) and the trail making test (TMT). Studies assessing quality of life mainly used the Parkinson’s disease questionnaire (PDQ-39), except one study using the Parkinson’s disease quality of life questionnaire (PDQL) [16]. Details regarding the intervention characteristics of each study are summarized in Table 2.

### 3.4. Effects of the Interventions on Motor Outcomes

The outcomes related to motor symptoms in this study for the meta-analysis include gait parameters (i.e., cadence, velocity, and stride length) and mobility.

#### 3.4.1. Cadence

Four studies involving 151 participants explored the effect of music-based interventions on cadence [26,27,55,56]. Figure 4a displays a forest plot of the mean differences and the 95% CI for the gait cadence based on the random effect model. The result showed no significant between-group differences (MD = 3.21, 95% CI = −4.15~10.57, *p* = 0.39). A relatively high level of heterogeneity was present (I^2^ = 76%), and the sensitivity analysis indicated that removing one study [26] can lower the level of heterogeneity (I^2^ = 32%) and change the statistical significance (MD = 6.50, 95% CI = 1.37~11.62, *p* = 0.01). 

#### 3.4.2. Velocity

Velocity of gait was evaluated in four studies involving 151 participants [26,27,55,56]. Figure 4b shows a forest plot of the mean differences and the 95% CI for the gait velocity on the fixed-effect model. The significant positive effect of music-based interventions on velocity and no heterogeneity among different studies was found (MD = 0.12, 95% CI = 0.07~0.16, *p* < 0.00001; I^2^ = 0%). These results indicate that music-based interventions could significantly increase the walking speed of patients with PD.

#### 3.4.3. Stride Length

A total of 151 participants were included in four studies on stride length [26,27,55,56]. Figure 4c demonstrates a forest plot of the mean differences and the 95% CI for the stride length on the fixed-effect model. The results showed a significant effect on stride length (MD = 0.04, 95% CI = 0.02~0.07, *p* < 0.002) and low heterogeneity (I^2^ = 22%), suggesting that music-based interventions might be beneficial to increase the stride length of patients with PD.

#### 3.4.4. Mobility

Four studies involving 125 participants examined the impact of music-based interventions on mobility measured by TUG [26,30,39,56]. TUG is a highly reliable and valid tool for assessing functional mobility in older adults, including balance and stability through observing getting up, walking, turning around, and sitting down [60,61]. Figure 4d displays a forest plot of the mean differences and the 95% CI for mobility on the fixed-effect model. This meta-analysis demonstrated that music-based interventions had a statistically significant effect on improving mobility in patients with PD (MD = −1.05, 95% CI = −1.53~−0.57, *p* < 0.0001) and no heterogeneity was found (I^2^ = 0%).

**Figure 4 ijerph-20-01046-f004:**
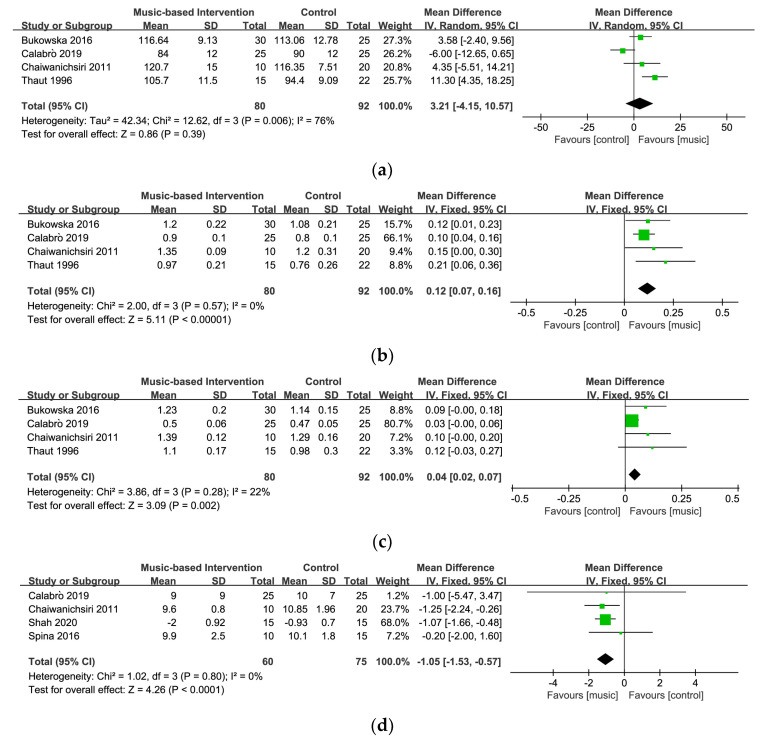
Forest plots for (**a**) cadence [26,27,55,56], (**b**) velocity [26,27,55,56], (**c**) stride length [26,27,55,56], and (**d**) mobility [26,30,39,56].

### 3.5. Effects of the Intervention on Non-Motor Outcomes

The outcomes of non-motor symptoms in this study for the meta-analysis included executive function (i.e., cognitive flexibility and inhibition) and quality of life.

#### 3.5.1. Cognitive Flexibility

Three studies with 82 participants measured cognitive flexibility using the TMT-part B [29,30,57]. Figure 5a displays a forest plot of the mean differences and the 95% CI for the cognitive flexibility on the random-effect model. The results showed no strong evidence of significant differences between the music-based intervention and the control groups (MD = 20.91, 95% CI = −10.62~52.44, *p* = 0.19). The analysis showed moderate heterogeneity across the studies (I^2^ = 66%). After the exclusion of one study [57], the level of heterogeneity was significantly reduced (I^2^ = 0%) and a significant result was found between the groups (MD = 39.43, 95% CI = 10.11~68.75, *p* = 0.008).

#### 3.5.2. Inhibition

Three studies with 82 participants utilized the cued color word Stroop test and the Korean Stroop test–color reading for the measurement of inhibition [29,30,57]. Figure 5b shows a forest plot of the standard mean differences and the 95% CI for the inhibition on the fixed-effect model. This meta-analysis demonstrated no statistically significant effect on inhibition (SMD = 0.07, 95% CI= −0.40~0.55, *p* = 0.76) and a relatively moderate heterogeneity level (I^2^ = 47%).

#### 3.5.3. Quality of Life

In five studies with 154 participants, four studies used the PDQ-39 [29,30,36,39] and one study used PDQL [16] for assessment of quality of life in the meta-analysis. Figure 5c displays a forest plot of the standard mean differences and the 95% CI for quality of life on the random-effect model. The results showed no evidence of significant differences between the music-based and control groups (SMD = −0.68, 95% CI = −1.68~0.32, *p* = 0.18) and a high level of heterogeneity (I^2^ = 88%). Removing one study [16] which used PDQL as the assessment tool significantly lowered the level of heterogeneity (I^2^ = 0%).

**Figure 5 ijerph-20-01046-f005:**
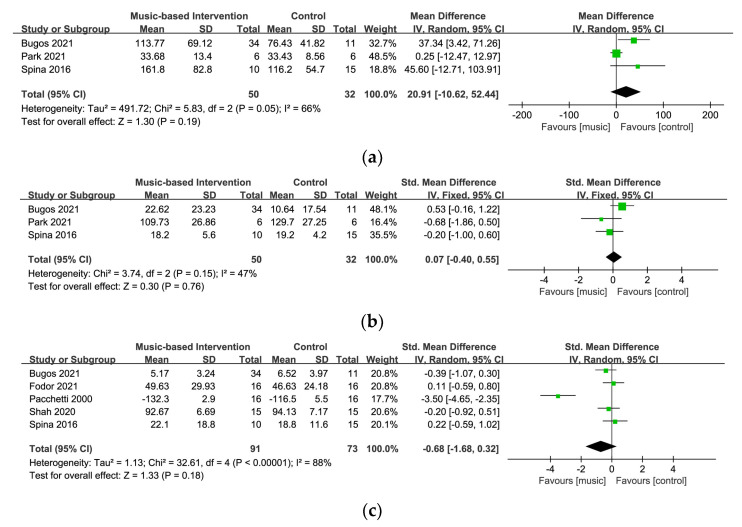
Forest plots for (**a**) cognitive flexibility [29,30,57], (**b**) inhibition [29,30,57], and (**c**) quality of life [16,29,30,36,39].

### 3.6. Sensitivity Analysis

First, a sensitivity analysis was performed to assess the impact of group allocation methods and the causes of heterogeneity. The exclusion of a CCT study [29] did not significantly affect the effect sizes nor the heterogeneity for the outcome of cognitive flexibility (MD = 13.63, 95% CI = –26.91~54.17, *p* = 0.51; I^2^ = 55%) and the outcome of quality of life (MD = −0.78, 95% CI = −2.12~0.56, *p* = 0.25; I^2^ = 91%). However, the outcome of inhibition after removing this CCT study showed a significant reduction in the level of heterogeneity (I^2^ = 0%). Additionally, a sensitivity analysis was conducted to verify whether the synthesized outcomes would change by excluding studies one-by-one. The results demonstrated few changes on the level of heterogeneity and effect size. For motor outcomes, the exclusion of a study [26] led to changes in the level of heterogeneity as well as the effect size (MD = 6.50, 95% CI = 1.37~11.62, *p* = 0.01; I^2^ = 32%) on cadence. Regarding non-motor outcomes, the exclusion of a study [57] was found to lead to significant changes in the heterogeneity as well as the effect size (MD = 39.43, 95% CI = 10.11~68.75, *p* = 0.008; I^2^ = 0%) on cognitive flexibility. For quality of life, after removing a study [16], the result led to a significant change in heterogeneity (SMD = −0.09, 95% CI = −0.45~0.27, *p* = 0.64; I^2^ = 0%). These results indicated that the evidence of the effect of music-based interventions on cadence, cognitive flexibility, and quality of life was not robust based on our meta-analysis. The complete sensitivity analysis is described in Appendix A.

### 3.7. Publication Bias

For the meta-analysis in this study, the researchers did not perform a publication bias assessment since the number of included studies per outcome was not enough to do so.

## 4. Discussion

This systematic review and meta-analysis investigated the therapeutic effects of music-based interventions in patients with PD. After a rigorous examination of the selected studies, researchers analyzed the outcomes of motor and non-motor symptoms in patients with PD. This review was distinguished from previous systematic reviews and meta-analyses by targeting both motor and non-motor symptoms in individuals with PD, applying active and passive musical activities as interventions. The results of this current review demonstrated the positive effects of music-based interventions on walking velocity, stride length, and mobility in individuals with PD. However, the results of the meta-analysis did not support the significant effects of music-based interventions on step cadence and other non-motor symptoms such as cognitive flexibility, inhibition, and quality of life.

On the basis of the outcomes in motor symptoms, the results support that music-based interventions could effectively improve gait-specific outcomes, including walking velocity and stride length. All of the original studies included in our meta-analysis showed a significant increase both in velocity and stride length [26,27,55,56]. These results are quite consistent with previous systematic reviews and meta-analysis studies which indicate that music-based interventions, specifically using auditory rhythmic cueing, appear to be promising therapeutic tools for the improvement of gait-related functions [9,43,44]. In addition, the present findings confirm that music-based interventions could also have a positive impact on functional mobility in individuals with PD. The original studies analyzed in a meta-analysis demonstrated significant results in improving mobility [26,39,56] and those studies provided the rhythmic element of music in order to facilitate participants’ walking functions. Regarding previous research which showed the significant effects of musical activities on mobility in people with PD, several meta-analyses have emphasized the therapeutic use of rhythmic cueing [9,44,45,46,47]. It has been widely discussed in plenty of published literature that rhythm-based interventions such as RAS can improve the spatiotemporal and kinematic features of gait including velocity, stride length, mobility, and balance in patients with PD [62]. Rhythmic auditory cues such as metronome beats or a strong beat embedded in music can regulate timing and pace in walking [63]. RAS may function as an external timekeeper that supports the impaired function of the defective basal ganglia in patients with PD [64,65], possibly through the involvement of compensatory networks in the brain [66]. Neuroimaging data may support this evidence, as the data suggest that alternative brain routes are activated during movement supported by RAS [67]. Therefore, music-based interventions, specifically using rhythmic auditory cues, may use scientific-evidence-based methods to optimize gait and gait-related functional mobility in patients with PD.

A trend toward increasing step cadence was observed in this review, but the between-group differences were not statistically significant. As for the sensitivity analysis for cadence, the exclusion of the study by Calabrò et al. [26] led to a reduction in the level of heterogeneity and a change of a significant effect size. It is necessary to consider that treadmill walking was used in this study, whereas free (ground) walking was used in the other studies [27,55,56] when evaluating cadence. In a systematic review and meta-analysis study on the gait parameters of PD patients, the evaluation method (free/ground walking vs. treadmill) was suggested as one of the factors influencing the cadence evaluation results [68]. The different environments of measuring cadence could, therefore, lead to different results. Another factor which may impact the differences between studies with regard to cadence may be explained by differences in the stimulation frequencies that were provided during the training sessions [67]. Two studies [26,56] in our meta-analysis provided gait training with a gradually increased frequency, whereas other studies [30,39] did not specify stimulation frequency. The provision of different stimulation frequencies for training may also influence the synthesized results of cadence. In this regard, future study needs to consider a standardized measurement environment or application of a consistent stimulation frequency for cadence outcomes. Additionally, the reason for the inconsistent outcome of cadence may be the difference in the different severities of PD. A previous study pointed out that step frequency is slightly reduced in the early stages of PD and as the disease progresses, the patient’s stride lengths decrease, while the step frequency increases in flustered gait [44]. Because of the different characteristics of step cadence according to the level of severity of PD, the results of cadence may be inconsistent.

As for the outcomes in non-motor symptoms, this review could not confirm the significant effect on executive functions such as cognitive flexibility and inhibition in patients with PD. Interestingly, all of the studies included in our meta-analysis for executive functions demonstrated significant effects of music-based interventions on both cognitive flexibility and inhibition and each study analyzed the results with the pre-post difference change values [29,30,57]. In other words, the degree of improvement in the experimental group was significantly greater than in the control group when compared with the differences between the pre- and post-results in each study. However, the use of comparison of post-intervention values is preferred in a meta-analysis and thus the method for calculating mean differences may contribute to the observed effect [67]. Music training can transfer to an extensive range of cognitive domains and may serve as an effective cognitive training intervention for patients with PD [69]. A recent study found intense piano training significantly enhanced executive functions in patients with PD as measured by the Stroop test and TMT [29]. These results have important implications for the therapeutic use of music-based interventions for cognitive functions. However, the evidence for the outcomes for cognitive functions remains too limited to draw any conclusions. More well-designed studies specifically focusing on executive functions in individuals with PD should be conducted.

Additionally, our meta-analysis results showed that the outcome of quality of life did not reach statistical significance. Some studies [16,19,30,39] have shown that music-based interventions can improve the quality of life of patients with PD, but another study [29] showed opposite results. A study by Bugos et al. [29] pointed out that an intensive period of intervention (10 days, 3 h per day) may cause a negative impact on quality of life. In this study, the participants were given intensive tasks because the intervention focused on improving executive function. As a result, executive function was improved, but the results for the quality-of-life test would have decreased due to the increase in fatigue. Therefore, the duration of training and rest between the sessions should be considered to optimize psychosocial outcomes. Several previous meta-analyses on quality of life have found no significant effect after interventions [9,45,46,47], and these results are in line with our findings. One of the possible reasons is that the interventions were geared towards the outcomes of physical movement or gait (e.g., movement, RAS, and treadmill training) and were associated with improvement in quality of life as a secondary outcome. Therefore, future studies should focus on carrying out interventions directly related to quality of life as a primary outcome in order to observe an improvement in psychosocial rehabilitation in patients with PD.

In cases where the effect size between the studies was inconsistent, a sensitivity analysis was conducted to examine the factors influencing the results. As a result of examining the effect of the group assignment method (random assignment or non-random assignment), it was confirmed that the inclusion of the CCT study [28] in the analysis did not significantly affect the results. Regarding measurement tools, similar but different scales were used in the quality-of-life evaluation, so SMD was used for analysis, but high heterogeneity occurred due to differences in the measurement values. In addition, in some studies, there was a possibility that the method for calculating mean differences using only the post-value may affect the analysis results [67]. It was confirmed that in the comparison between the groups that the comparison of the pre-post change values and the comparison of only the post-values can show different results especially in non-motor outcomes.

In the risk-of-bias evaluation, the results were reflected in this current study selection to minimize the risk of bias in the analysis results. Among the RCT studies, 4 out of 14 studies were excluded from the meta-analysis due to a high risk of bias and all 3 CCT studies were included in the analysis because none of the 3 CCT studies were at critical risk. Four studies were evaluated as high risk in missing outcome data and selectively reporting outcomes, suggesting that issues from these factors needs to be considered in future studies [19,38,58,59]. Another major common cause of increasing the risk of bias was the failure to blind participants in the group allocation and/or assessment process. First, several studies did not report the specific methods of the randomization process. In most trials, allocation concealment and the blinding allocation process were not possible due to the nature of the music-related activities; however, if blinding is not achieved, the true effect of interventions cannot be estimated [70]. In addition, using a blinding assessment process, especially with subjective measures, was not possible in most music-based studies. For example, quality of life can only be measured subjectively as rated by a patient. Outcome assessors may be biased towards the experimental group and rate their observations more positively to affect the outcomes in favor of the music-based treatment. Such bias could lead to inflation of the treatment effect size [70]. Further studies should ensure appropriate methods which can control biases from randomization, blinding, selection, and measurement outcomes.

This review has certain limitations. First, the sample size included in the meta-analysis was relatively small and varied between the studies, which may have affected the quality of the evidence, and therefore, future studies with a larger sample size are necessary to draw reliable results. Second, the literature included only publications written in English, which may increase the risk of publication bias. In addition, only 11 studies contributed to the meta-analysis in this review, which could not be conducted for all outcomes nor all combinations of interventions and comparators. Therefore, future research and evidence syntheses should include a wider range of outcomes relating to the potential benefit of various types of music-base interventions. Those outcomes could include motor symptoms, including gait freezes and falls and upper extremity and fine motor rehabilitation, as well as non-motor symptoms, including communication and language and cognitive and psychosocial rehabilitation in patients with PD. In addition, owing to the insufficient number of trials included in each outcome, the confirmation of publication bias had a limited aspect. Lastly, there was limitation in conducting subgroup analyses on various factors including the type of interventions and the severity of disease due to the limited number of included studies. It would be necessary to carry out a multilateral analysis of the variables that influence the results in the future. These limitations may have affected the evaluated outcomes for motor and non-motor symptoms in patients with PD.

## 5. Conclusions

This systematic review and meta-analysis confirmed the evidence that music-based interventions can have a positive effect on motor symptoms in patients with PD. However, there was insufficient data to fully confirm the effect of music-based interventions on non-motor symptoms and the results should be interpreted with some caution. Future studies on non-motor symptoms in individuals with PD need to be more actively conducted and well-designed studies with larger sample sizes are necessary to draw definite conclusions on music-based interventions for motor and non-motor symptoms in individuals with PD. In addition, rigorous randomized controlled trials (RCTs) that can reduce the heterogeneity of the results need to be considered.

## Figures and Tables

**Figure 1 ijerph-20-01046-f001:**
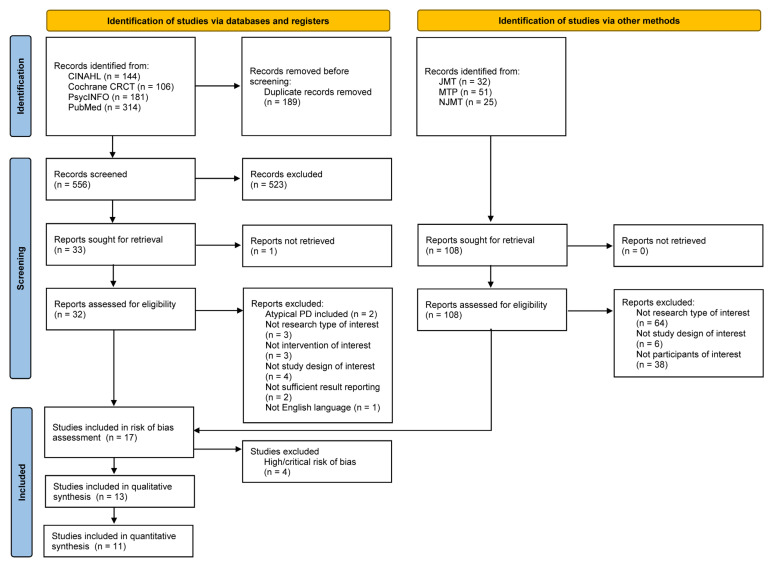
Flow diagram of the study selection process according to the preferred reporting items for systematic reviews and meta-analyses (PRISMA) 2020 Guideline.

**Figure 2 ijerph-20-01046-f002:**
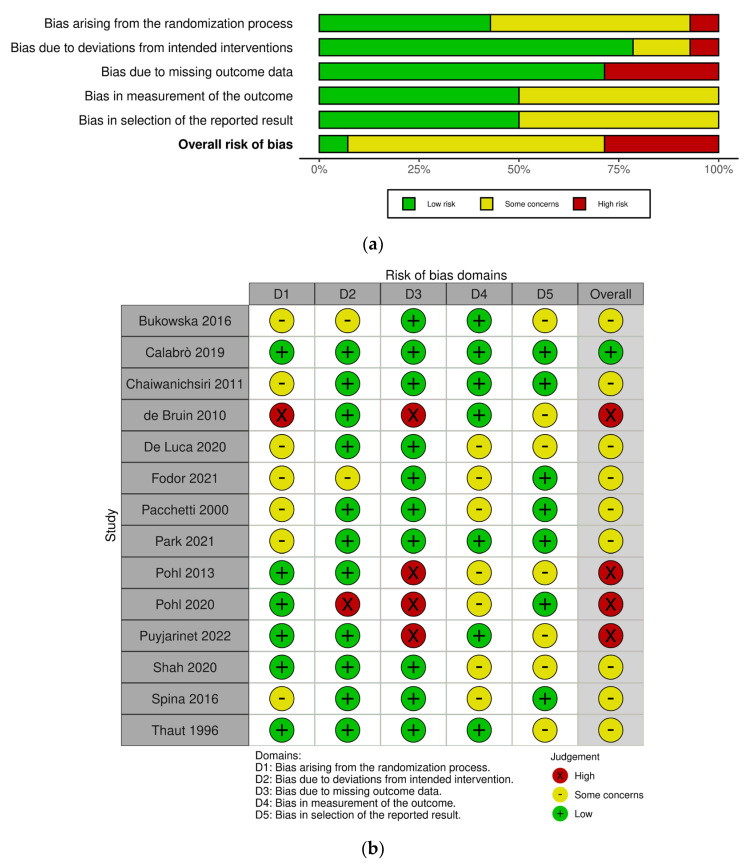
Risk-of-bias graphs for the randomized controlled trial (RCT) studies. (**a**) Risk of bias as a percentage and (**b**) risk-of-bias summary [16,19,26,27,30,36,38,39,54,55,56,57,58,59].

**Figure 3 ijerph-20-01046-f003:**
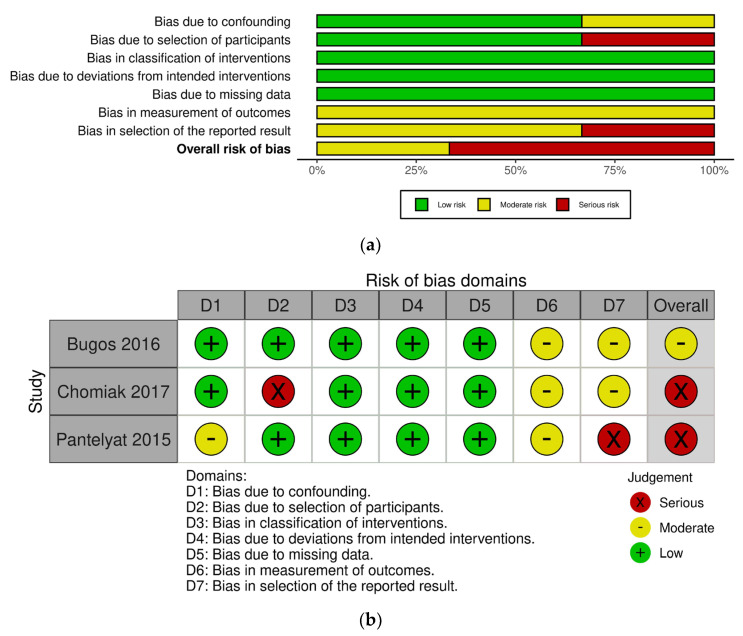
Risk-of-bias graphs for the controlled clinical trial (CCT) studies. (**a**) Risk of bias as a percentage and (**b**) risk-of-bias summary [28,29,37].

**Table 1 ijerph-20-01046-t001:** Characteristics of participants.

First Author	Country	Study	Sample		Age		Gender		Disease Duration (Years)		UPDRS		Hoehn and Yahr Score	
(Year)		Types	Ex	Co	Ex	Co	Ex	Co	Ex	Co	Ex	Co	Ex	Co
Bugos (2021) [29]	USA	CCT	34	11	65.79 ± 8.38	67.55 ± 7.29	M11/F18	M6/F5	N/A	N/A	N/A	N/A	1 or 2 ^a^	1 or 2 ^b^
Bukowska (2016) [27]	Poland	RCT	30	25	63.4 ± 10.61	63.44 ± 9.67	M15/F15	M15/F10	N/A	N/A	N/A	N/A	2 or 3	2 or 3
Calabrò (2019) [26]	Italy	RCT	25	25	70 ± 8	73 ± 8	M11/F9	M14/F6	10 ± 3	9.3 ± 3	29 ± 3	31 ± 5	3 ± 1	3 ± 1
Chaiwanichsiri (2011) [56]	Thailand	RCT	10	10/ 10	67.1 ± 4.0	67.9 ± 6.3/ 68.6 ± 5.2	N/A	N/A	3.7 ± 4.1	7.4 ± 3.4 4.4 ± 2.3	N/A	N/A	2, 2.5, 3	2, 2.5, 3
Chomiak (2017) [28]	Canada	CCT	5	6	70.8 ± 5.6	69.0 ± 5.7	M5/F0	M4/F2	15.4 ± 5.4	11.2 ± 5.0	18.2 ± 6.4 (Part III)	20.3 ± 4.7 (Part III)	2.5 ± 0.50	2.7 ± 0.41
De Luca (2020) [54]	Italy	RCT	20	20	63.2 ± 8.4	66.5 ± 6.2	M10/F10	M7/F13	N/A	N/A	N/A	N/A	1.5 ± 0.53	1.7 ± 0.59
Fodor (2021) [36]	Romania	RCT	16	16	67.1 ± 5.9	65.6 ± 5.5	M4/F12	M6/F10	N/A	N/A	N/A	N/A	1~3	1~3
Pacchetti (2000) [16]	Italy	RCT	16	16	62.5 ± 5	63.2 ± 5	M12/F4	M11/F5	4.8 ± 3	5.2 ± 2	40.2 ± 7.7 (Part III)	40.7 ± 7 (Part III)	2 or 3	2 or 3
Pantelyat (2016) [37]	USA	CCT	8	10	72 ± 8	71 ± 7	F% 63	F% 60	N/A	N/A	22 ± 6 (MDS Part III)	25 ± 11 (MDS Part III)	2.2 ± 0.46	2.2 ± 0.24
Park (2021) [57]	Korea	RCT	6	6	61.6 ± 4.9	63.1 ± 10.1	N/A	N/A	5.6 ± 3.1	4.8 ± 1.4	N/A	N/A	2 or 3	2 or 3
Shah (2020) [39]	India	RCT	15	15	66.98 ± 5.4	68.78 ± 4.76	M11/F4	M12/F3	2 ± 1.51	3 ± 1.58	N/A	N/A	2, 2.5, 3 (modified)	2, 2.5, 3 (modified)
Spina (2016) [30]	Italy	RCT	10	15	68.3 ± 8.1	61.9 ± 6.8	N/A	N/A	79.2 ± 39.2 (months)	80 ± 53.4 (months)	18.5 ± 8.1 (MDS Part III)	17.7 ± 8.3 (MDS Part III)	N/A	N/A
Thaut (1996) [55]	U.S.A.	RCT	15	11/ 11	69 ± 8	74 ± 3/ 71 ± 8	M10/F5	M8/F3 M8/F3	7.2 ± 4	5.4 ± 3/ 8.5 ± 4	N/A	N/A	2, 2.5, 3 (modified)	2, 2.5, 3 (modified)

Note: CCT = controlled clinical trials; Co = control group; Ex = experimental group; F = female; M = male; MDS = The Movement Disorder Society-sponsored revision of the UPDRS; RCT = randomized controlled trials; UPDRS = unified Parkinson’s disease rating scale; ^a b^ stages 1 or 2 of disease as measured by the UPDRS.

**Table 2 ijerph-20-01046-t002:** Characteristics of interventions.

First Author (Year)	Contents	Intervention		Control	Outcomes	Evaluation Methods
		Intensity (Duration; Frequency; Time)	Type			
Bugos (2021) [29]	Intensive piano training	10 days; daily each morning; 3 h/session	Group	No treatment	⑦⑨⑩	PASAT, cued color–word Stroop, digit coding, symbol search, TMT part B, D-KEFS: verbal fluency subtest; PDQ-39; GSE
Bukowska (2016) [27]	The combination of three NMT sensorimotor techniques: TIMP, PSE, and RAS for daily life activities, balance, pre-gait, and gait pattern	4 weeks; 4 times/week; 45 min/session	Individual	Stay active and perform daily life activities b/t measures	①②	Computerized dynamic posturography CQ stab; optoelectric 3D movement analysis, system BTS Smart
Calabrò (2019) [26]	Treadmill gait training using GaitTrainer3 with RAS	8 weeks; 5 times/week; 20 min/session	Individual	Treadmill gait training using GaitTrainer3 without RAS (same intensity)	①②⑩	UPDRS, BBS, TUG; A single wireless inertial sensor (GSensor, BTS Bioengineering), 10 m walking test, FGA, gait quality index; Tinetti FES
Chaiwanichsiri (2011) [56]	Treadmill with music cue 3 days/week and home walking program 3 days/week	8 weeks; 6 days/week; 30 min/session	Individual	Control 1. Treadmill training 3 days/week and home walking program 3 days/week Control 2. Home walking program 6 days/week	①②	TUG, single leg stance test; expanded TUG, six-minute walk test
Chomiak (2017) [28]	In-home music-contingent stepping-in-place training	4 weeks; minimum of 3/week; 10 to 20 min/session	Individual	In-home auditory-contingent stepping-in-place training using a podcast (auditory feedback) (same intensity)	①⑧⑩	FOG-Q; MoCA; FES-International
De Luca (2020) [54]	Gait training via a treadmill and RAS	8 weeks; 3 times/week; about 30 min/session	Individual	Traditional over-ground gait training (same intensity)	①②⑧ ⑨⑩	FIM: motor, TUG; 10 m walking test; FIM: cognitive; PGWBI; Brief COPE
Fodor (2021) [36]	A multimodal rehabilitation program centered on physical therapy combined with listening to music	14 days; daily; 2.5 h	Group	A multimodal rehabilitation program centered on physical therapy (same intensity)	⑨	PDQ-39
Pacchetti (2000) [16]	Music therapy, choral singing, voice exercise, rhythmic and free body movements, and improvisation	13 weeks; weekly; 2 h/session	Group	PT, stretching exercises, specific motor tasks, and balance and gait training (intensity: 13 weeks; weekly; 1.5 h/session)	①⑨⑩	UPDRS-ADL, UPDRS-MS; PDQL; happiness measure
Pantelyat (2016) [37]	West African drum circle	6 weeks; twice-weekly; 45–60 min/session	Group	No treatment	⑧⑨⑩	MoCA; PDQ-39; GDS-15
Park (2021) [57]	Drum playing with rhythmic cueing (DPRC intervention)	12 weeks; weekly; 50 min/session	Group	Regular program: gait rehabilitation and speech therapy	①③⑦	Nine-hole peg test; K-TMT-e: part A; K-TMT-e: part B, Korean Stroop test
Shah (2020) [39]	PT with MT: standing and marching activity, gait training, tap dance, and balance training	6 weeks; 4 times per week; 60 min/session	Group	Conventional PT	①②⑨	TUG; dynamic gait index; PDQ-39
Spina (2016) [30]	Active music therapy—production of music, singing, and dancing	24 weeks; once a week; 90 min/session	Group	No treatment	①③④⑤⑥⑦⑨	New FOG-Q, TUG, MDS-UPDRS; TMT part A; Rey immediate recall, Rey delayed recall; clock drawing test; names denomination, verbal denomination; FAB, phonemic verbal fluency, Raven test, TMT part B, Stroop test; PDQ-39
Thaut (1996) [55]	RAS; home-based training	3 weeks; daily; 30 min/session	Individual	Control 1. Self (internally)-paced group: same exercise program but without the aid of RAS Control 2. No-training group: normal daily activities	①②	EMG; computerized foot-switch recording system

Note. ① = General motor function; ② = gait parameters; ③ = attention; ④ = memory; ⑤ = visuospatial function; ⑥ = language; ⑦ = executive function; ⑧ = general cognitive function; ⑨ = quality of life; ⑩ = mental health; BBS = Berg balance scale; Brief COPE = brief coping orientation to problems experiences; D-KEFS = Delius–Kaplan executive function system; EMG = electromyography; FAB = frontal assessment battery; FES = falls efficacy scale; FGA = functional gait assessment; FIM = functional independence measure; FOG-Q = freezing of gait questionnaire; GDS-15 = geriatric depression scale-15; GSE = general self-efficacy; K-TMT-e = Korean trail making test for the elderly; MoCA = Montreal cognitive assessment; PASAT = paced auditory serial addition test; MT = music therapy; PDQ-39 = Parkinson’s disease questionnaire-39; PDQL = Parkinson’s disease quality of life questionnaire; PGWBI = psychological general well-being index; PT = physical therapy; RAS = rhythmic auditory stimulation; TMT = trail making test; TUG = timed up-and-go test.

## Data Availability

All relevant data are included in the study and Appendix A.

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
