# Peer review of "Effects of Music-Based Interventions on Motor and Non-Motor Symptoms in Patients with Parkinson’s Disease: A Systematic Review and Meta-Analysis"

_ijerph, 2023, doi:10.3390/ijerph20021046_

Round 1

Reviewer 1 Report

I appreciate the researchers’ effort to present ‘Effects of music-based interventions on motor and non-motor symptoms in patients with Parkinson’s disease: A systematic review and meta-analysis’. However, there are lack of the purpose of systematic review and meta-analysis for Parkinson’s disease patients without UPDRS and Hohen & Yahr scale information.

A systematic review is a well-planned literature review that basically answers a focused research problem, with pre-defined inclusion and exclusion criteria. Meta analysis is a statistical analysis that consists of huge collections of outcomes for the purpose of integrating the findings based on pre-defined inclusion and exclusion too.

Unified Parkinson's Disease Rating Scale (UPDRS) is a rating tool used to gauge the the severity and progression of Parkinson's disease in patients. The UPDRS scale consists of the following six segments: 1) Mentation, Behavior, and Mood, 2) ADL, 3) Motor sections, 4) Complications of Therapy (in the past week) 5) Modified Hoehn and Yahr Scale, 6) Schwab and England ADL scale. The first four segments are made up of 42 items grouped into four subscales. The UPDRS was developed in 1987 as a gold standard by neurologists for monitoring the response to medication used to decrease the signs and symptoms of Parkinson's. The Hoehn and Yahr scale is a commonly used system for describing how the symptoms of Parkinson's disease progress.  It included stages 1 through 5. Since then, a modified Hoehn and Yahr scale was proposed with the addition of stages 1.5 and 2.5 to help describe the intermediate course of the disease.

It is somewhat lacking in research significance to conduct a systematic and meta-analysis by selecting studies that do not provide important clinical information such as UPDRS and Hohen & Yahr scale on patients with Parkinson's disease to verify the effectiveness of interventions in medical and clinical research. It means This means that articles in which interventions were provided in a state where the level of patients with various symptoms were not accurately considered clinically were included somewhat. That’s why the previous several review studies for the effects of music-based interventions in PD were introduced only potential benefits by various interventions due to lack of UPDRS and H & Y scale.

Among the studies selected for this study, we hope to present the results in accordance with the purpose of this study by excluding studies for which UPDRS and H & Y information on Parkinson's disease patients were not provided.

Reviewer 2 Report

Dear authors

This systematic review and meta-analysis reports the influence of rhythmic-based interventions on both moto and non-motor symptoms in PD. Reporting both motor and non-motor symptoms is an added value in this research topic in PD. Although the introduction is comprehensive and the aim is clear and the risk of bias is performed precise. However there are still some concerns and questions.

1.    The article reviews music-based intervention in PD. However, in the inclusion and exclusion criteria it is mentioned that studies using movement therapy with music and dance are excluded. However, studies using metronomes (RAS) are included. Can the authors specify “music-therapy” in their introduction or method of inclusion?

In the introduction the authors cite a recently published systematic review and meta-analysis that reported the influence of RAS on gait in PD, namely the systematic review of Wang et al. (2022) (reference 44). The systematic review of Wang et al .(2022) included 19 studies in the systematic review and 12 in the meta-analysis. The amount of included studies is therefore higher in the study of Wang et al. (2022) compared to this review (5 studies using RAS). Is it possible that this review missed some relevant studies by using only “Music” in the search strategy?

The participant characteristics are summarized in table 1. Freezing of gait is a frequently observed symptom in PD, however this is not reported in the table. The influence of music- or rhythm-based interventions will differ accordingly as the authors already describe for disease severity. Can the authors specify if the participants have freezing of gait in table 1?

The results are clearly described and important critical thoughts are reflected in the discussion. Improvements motor-symptoms are discussed, however can the authors describe the results also in view of the clinical important change ?

Thank you for addressing these questions and concerns.

Reviewer 3 Report

 In their manuscript entitled “Effects of Music-based Interventions on Motor and Non-motor Symptoms in Patients with Parkinson’s Disease: A Systematic Review and Meta-Analysis” (manuscript-ID ijerph-2101011), Hyunjung Lee and Bumsuk Ko present, as it is already described in the title, a review and meta-analysis on effects of music-based interventions on motor and non-motor symptoms in subjects with Parkinson´s disease including 13 studies involving 417 participants describing randomized controlled and controlled clinical trials. They found with high significance that music-based interventions can improve walking velocity, stride length, and mobility. On the other hand, they did not find significant effects of music-based interventions on cadence, cognitive flexibility, inhibition, and quality of life. The authors conclude that music-based interventions are effective for the improvement of some motor, but not non-motor symptoms.

In my opinion, the authors did a good job on a very relevant topic. The choice of included studies including the method of search and selection as well as their analysis seems appropriate and the manuscript is well written in all ist parts including tables and figures.

 The analysis was done seperately for different outcomes within the different motor and non-motor symptoms. Although - and that is my main concern - it seems very ambitious to calculate for such a heterogeneous disease as Parkinson's disease a meta-analysis for such different interventions with different control groups and assessment tools. However, this manuscript offers a good overview of the various intervention options and their effects, which has so far been done in separate individual reviews. And the authors draw the right conclusion: that future studies on non-motor symptoms in PD need to be more actively conducted in well-designed studies with larger sample sizes in rigorous randomized controlled trials to reduce the heterogeneity of results.

Round 2

Reviewer 1 Report

Thanks to the authors for quickly correcting my comments. However, it seems that the authors still lack an understanding of the exact clinical conditions of the selected subjects to verify the effectiveness of clinical diagnosis and intervention of Parkinson's disease. Also, since music intervention is provided to groups, not individuals, please clearly modify the title to group music intervention

 In detail, I will explain again to help you understand the target group selection conditions. Prior to 2010, it was not easy to conduct clinical researches on non-pharmacological treatment for Parkinson's disease patients, so there was less accurate clinical diagnosis of Parkinson's disease patients, that is, UPDRS and H&Y scale information, or the study was conducted by recruiting a target group only with H&Y scale information. Therefore, the verification of the mediation effect may has not been effectively conducted. 

  However, since the last 10 years or so, in all credible clinical studies on Parkinson's disease patients in medical settings, if there is no information on the UPDRS and H&Y scale in the selection criteria for Parkinson's disease patients, they are not recruited into the target group at all. Do the authors know exactly about the clinical diagnosis process of Parkinson's disease patients in the hospital? The diagnosis of Parkinson's disease in the hospital is performed by a neurologist, and the UPDRS and H&Y scale information I mentioned are not selected by the authors arbitrarily, but both diagnostic information is required. 

  Therefore, please select an article that provides both UPDRS and H&Y scale information among a total of 13 selected articles and analyze it again to verify the effectiveness of group music interventionThank you.
